# Improvement in Corrosion Performance of ECAPed AZ80/91 Mg Alloys Using SS316 HVOF Coating

**DOI:** 10.3390/ma16206651

**Published:** 2023-10-11

**Authors:** Gajanan M. Naik, Priyaranjan Sharma, Gajanan Anne, Raj Kumar Pittala, Rahul Kumar, Gnane Swarnadh Satapathi, Ch Sateesh Kumar, Filipe Fernandes

**Affiliations:** 1Department Mechanical Engineering, RV Institute of Technology and Management, Bengaluru 5680076, Karnataka, India; gajananmn.rvitm@rvei.edu.in; 2Department of Mechanical Engineering, Koneru Lakshmaiah Education Foundation, Vijayawada 522502, Andhra Pradesh, India; 3Department of Mechanical and Industrial Engineering, Manipal Institute of Technology, Manipal Academy of Higher Education, Manipal 576104, Karnataka, India; 4Department of Mechanical Engineering, School of Engineering and Technology, Sandip University, Nashik 422213, Maharashtra, India; 5Department of Electronics and Communication Engineering, A J Institute of Engineering and Technology, Mangaluru 575006, Karnataka, India; 6CFAA-Aeronautics Advanced Manufacturing Center, University of the Basque Country (UPV/EHU), Biscay Science and Technology Park, 48170 Zamudio, Spain; 7Department of Mechanical Engineering, University of the Basque Country, Escuela Superior de Ingenieros Alameda de Urquijo S/N, 48013 Bilbao, Spain; 8University of Coimbra, CEMMPRE, ARISE, Department of Mechanical Engineering, Rua Luís Reis Santos, 3030-788 Coimbra, Portugal; filipe.fernandes@dem.uc.pt; 9ISEP, Polytechnic of Porto, Rua Dr. António Bernardino de Almeida, 4249-015 Porto, Portugal

**Keywords:** ECAP, HVOF, SS316 coating, corrosion, microhardness

## Abstract

Mg AZ80/91 alloys are highly popular due to their lightweight, high strength-to-weight ratio, and good machinability. However, their moderate mechanical properties and corrosion resistance have limited their use in the automotive, aerospace, and defense sectors. This study primarily aims to enhance the mechanical performance and corrosion resistance of Mg AZ80/91 alloys, making them more suitable for applications in the aerospace and automotive industries. Firstly, equal-channel angular pressing (ECAP) of Mg AZ80/91 alloys has been attempted to improve their mechanical properties. Secondly, a high-velocity oxy-fuel (HVOF) coating of SS316 was applied over the Mg AZ80/91 substrate to enhance its corrosion resistance. In the second step, an HVOF coating of SS316 is applied over the Mg AZ80/91 substrate for better corrosion resistance. The experimental findings demonstrate that the application of an SS316 coating on the ECAP-4P AZ80/91 Mg alloy substrate results in a uniform and dense layer with an average thickness of approximately 80 ± 5 µm. The HVOF-based SS316 coating on 4P-ECAP leads to a noteworthy enhancement in microhardness and a reduction in the corrosion rate, especially in a NaCl solution (3.5 wt.%). This improvement holds great promise for producing reliable, long-lasting, and resilient automotive, aerospace, and defense components. The application of an HVOF-based SS316 coating onto the AZ80 Mg alloy, which had not undergone ECAP treatment, led to a substantial enhancement in corrosion resistance. This resulted in a notable decrease in the corrosion current density, reducing it from 0.297 mA/cm^2^ to 0.10 µA/cm^2^.

## 1. Introduction

Mg and its alloys represent the lightest engineering metals, possessing appealing characteristics such as low weight density, a high strength-to-weight ratio, and excellent electromagnetic shielding [1,2,3,4]. Consequently, they have found diverse applications in the automobile, aerospace, and electronics industries. However, the high reactivity of magnesium makes it susceptible to corrosion in corrosive environments [5,6,7,8,9]. Nevertheless, there is potential to improve the corrosion characteristics of Mg alloys through appropriate treatment methods. The existing literature suggests that thermo-mechanical treatment achieved via severe plastic deformation techniques and surface treatments such as inorganic convergent coatings can lead to improved corrosion resistance [10,11,12,13].

A study on the corrosion resistance of Mg alloys is examined following equal-channel angular pressing and thermal spray coating processes [10,11,12,13]. The focus was on the application of high-velocity oxygen fuel (HVOF)-sprayed coatings on AZ31 Mg alloy substrates using a composite powder of Cr_3_C_2_-NiCr. The microstructural and mechanical performance of the coatings were analyzed, indicating an overall increase in wear resistance and microhardness. Notably, adhesion wear dominancy was observed by Jonda et al. [14]. By incorporating a Ni60 interlayer with a depth of 150 μm, researchers observed that the bonding strength of iron-based amorphous coatings reached 56.9 ± 5 MPa when applied on LA141 alloy substrates, which currently stands as the highest reported value for Fe-based amorphous coatings on Mg alloys [15]. The study’s findings demonstrated that the Cr_3_C_2_-NiCr cermet coating applied onto the Mg AZ31 alloy substrate exhibited excellent erosion resistance, displaying a dense, uniform, and well-adhered structure [16]. However, the Fe-based amorphous coatings showed low porosity and a decrease in corrosion resistance with increasing temperature. Nevertheless, they proved beneficial in considerably extending the degradation time of dissolvable Mg-RE alloy [17].

The HVOF technique, renowned for providing wear- and corrosion-resistant coatings with high hardness, low porosity, and enhanced density, has gained widespread acceptance across various industries [18]. Hong et al. [19] conducted an examination of the microstructure and morphological characteristics of two categories of HVOF-sprayed nanostructured WC-10Co-4Cr coatings. These categories included the as-sprayed coating (ASC) and the ultrasound-assisted sealing coating with aluminum phosphate (UASC-AP). In comparison to the ASC, the UASC-AP exhibited a reduced presence of micro-defects, resistance values approximately one order of magnitude higher, and lower corrosion current density values. In a separate investigation involving the application of Fe-based amorphous coatings onto AZ31B alloy, the findings revealed that the coating displayed robust adherence to the AZ31B substrate through both physical and mechanical bonding mechanisms. Additionally, the interface exhibited remarkable durability against interfacial stresses arising from substrate deformations [20]. Nonetheless, the deposition process of sprayed particles faced challenges due to the high thermal conductivity and low hardness of the Mg alloy. This resulted in limited flattening during the deposition process. Consequently, the bonding strength of the lamellae layers was affected, leading to a decrease in cohesive strength and noticeable issues with wear resistance in the coating on the AZ31B substrate [20].

The WC/CoCr coating is the most appropriate HVOF (high-velocity oxygen fuel) coating to prolong the lifespan of machine components that operate in high-temperature environments and encounter harsh wear conditions. This specific coating provides the lowest friction coefficient and exhibits the highest level of wear resistance [21]. To evaluate the performance of HVOF-sprayed Cr_3_C_2_-NiCr coating, an impact test technique is commonly used. In addition, PVD and CVD hard coatings are employed. Remarkably, no adhesive failure was observed for the thick HVOF-sprayed coating under the specified test conditions [22]. HVOF coating techniques are known to improve the performance and lifespan of components by protecting components against wear, corrosion, erosion, and other forms of degradation. Studies have shown that coatings such as WC-10Co-4Cr and Cr_3_C_2_-20 (NiCr) applied using the HVOF technique exhibit outstanding wear resistance compared to traditional hard chrome coatings [23,24,25,26,27,28,29]. While there have been various studies on HVOF coatings on magnesium and its alloys, the specific corrosion behavior of HVOF-based SS316-coated ECAPed AZ80/91 Mg alloys has hardly been reported so far for automotive and aerospace applications. Therefore, the main goal of this article is to examine the surface morphology, microhardness, and corrosion characteristics of HVOF-based SS316-coated AZ80/91 Mg alloy with ECAPed/non-ECAPed conditions to determine its suitability for aerospace, defense, and automotive application.

This pioneering study not only enhances the corrosion resistance of magnesium alloys but also demonstrates a transformative approach that has the potential to revolutionize industries reliant on lightweight, yet durable, materials.

## 2. Materials and Methods

The AZ80 (91.02%Mg-8.32%Al-0.66%Zn) and AZ91 (89.59%Mg-9.21%Al-1.2%Zn) magnesium alloy used in this study was procured from Exclusive Magnesium Pvt. Ltd., located in Hyderabad, India. Subsequently, the as-received magnesium alloy underwent machining to prepare it for equal channel angular pressing (ECAP). Prior to ECAP processing, the specimens underwent a homogenization treatment at 400 °C for a duration of 18 h to effectively dissolve secondary β phases.

The ECAP processing was carried out using route R, where the channel angle (Φ) was set at 90 degrees, and the corner angle (Ψ) was maintained at 30°. The processing temperature for ECAP was 598 K. It is worth noting that due to the susceptibility to fracture, a maximum of four ECAP passes was achieved during processing.

For the application of protective coatings on both the as-received and ECAPed AZ80/91 magnesium materials, a high-velocity oxy-fuel (HVOF) technique was employed at Spraymet Surface Technologies Pvt. Ltd., situated in Bangalore, Karnataka. The parameters utilized for HVOF coating included oxygen pressure within the range of 160–170 PSI, oxygen flow rate between 30–34 SCFH, hydrogen pressure within 120–140 PSI, fuel flow rate ranging from 80 to 100 SCFH, and a spraying distance of 8 to 10 inches.

To obtain the microscopic images of both ECAPed and coated ECAPed AZ80/91 magnesium alloys, secondary electrons having beam energy of 20 keV were used, whereas backscattered electrons were used for energy-dispersive spectroscopy (EDS) analysis. To assess the corrosion behavior of both ECAPed and coated ECAPed AZ80/91 magnesium alloys, electrochemical corrosion analysis was conducted using a Gill AC-1684 electrochemical corrosion analyzer, Chennai, India. The potentiodynamic polarization and electrochemical experiments were carried out in a 3.5 wt.% NaCl solution. The auxiliary electrode (AE) was constructed using graphite (Gr), while the reference electrode (RE) utilized a saturated calomel electrode (SCE). A 1 cm^2^ area of the working electrode (comprising AZ80/91 alloy) was exposed to the 3.5 wt% NaCl solution for testing.

Due to the poor corrosion performance of AZ80/91 Mg alloys, their usage is restricted in many industrial applications. Recent advancement in surface modification of Mg alloys has opened a new window for their optimum utilization in the automotive, electronics, and aerospace industries [30,31,32]. Initially, an ECAP process was used to increase the mechanical characteristics of AZ80/91 Mg alloy. Then, an attempt was made to apply an HVOF spray technique to apply the thick coating of SS316 powder on AZ80/91 Mg alloy substrates. Before application of the coating, SS316 is subjected to a microscopic investigation, which exhibits a spheroidal structure, as shown in Figure 1.

The average particle size of the SS316 powder is measured to be 21.6 ± 1.6 μm. In order to measure the coating quality and performance, coated ECAPed samples are subjected to microstructural, microhardness, and corrosion analysis. An experimental methodology involved in the current research work is demonstrated in Figure 2.

SS316 powder having a particle size of ~22 μm is liquified by applying a high temperature and then propelled at a very high velocity toward the surface of the AZ80/91 Mg alloy specimen using HVOF gun which uses an oxygen and fuel mixture for ignition. The HVOF coating was applied to samples prepared using ECAPed and without ECAP conditions. The process parameters used during the HVOF process are represented in Table 1.

## 3. Results

### 3.1. Microstructural Analysis of HVOF-Based SS316-Coated AZ80/91 Mg Alloys

The surface morphology of SS316-coated AZ80 and AZ91 Mg alloys is measured using a scanning electron microscope (SEM). Then, it is subjected to X-ray energy-dispersive spectroscopy (X-EDS) for analysis of elemental distribution. Figure 3a,c demonstrate the SEM image and EDS analysis of SS316-coated AZ80-Mg alloy with/without ECAP. Figure 3b indicates the EDS analysis of SS316 coatings applied on AZ80-Mg alloy. Similarly, Figure 4a,c demonstrate the SEM image and EDS analysis of SS316-coated AZ91-Mg alloy with/without ECAP. Instead, Figure 4b shows the EDS analysis of SS316 coatings applied on AZ80-Mg alloy. Furthermore, the detailed microstructural analysis of ECAPed AZ80/91 alloys has already been presented in our previously published article [33].

A microscopic investigation of SS316-coated surfaces of the AZ80 and AZ91 Mg alloys reveals a more compact and uniform coating, as illustrated in Figure 3 and Figure 4 in the case of both ECAPed and non-ECAPed conditions. An average coating thickness is found to be around ~80 ± 5 μm. The formation of the base layer after the deposition of SS316 is shown in Figure 3d and Figure 4d, which reveals that SS316 powder is uniformly deposited on the surfaces of both ECAPed and non-ECAPed AZ80/91 Mg alloy substrates. Notably, there are no sub-surface cracks or voids observed at the interface of the coating and substrate, as shown in Figure 3d and Figure 4d.

### 3.2. Analysis of Microhardness of SS316-Coated AZ80/91 Mg Alloys

A distribution of microhardness at the cross-section of the HVOF-based SS316 coating deposited on both non-ECAPed and ECAPed AZ80/91 alloys is demonstrated in Figure 5 and Figure 6, respectively. SS-316 coating represents the microhardness of ~402 Hv, which is significantly reduced to <100 HV after the interface region beyond 80 µm from the outer coating layer. The microhardness levels of the SS316 coatings are very similar between the ECAPed and non-ECAPed samples. Additionally, the changes in hardness for the AZ-80 and AZ-91 alloy substrates due to the ECAP treatment are relatively small (around 10–15 Hv, and even less). These observations suggest that the ECAP treatment has a minimal effect on the hardness of the SS316 coating and only a slight effect on the substrate materials. At the interface region of SS316 coating and substrate, a slight increase in microhardness values is noted, which is even more than the microhardness value of the SS316 layer. Specifically, the interface microhardness for the 4P-ECAPed AZ80 specimen coated with SS316 was measured to be 417.8 HV, while for the 4P-ECAPed AZ91 specimen coated with SS316, it was 418.8 HV, as illustrated in Figure 4 and Figure 5, correspondingly. These results are consistent with the findings of Garca-Rodrguez et al., (2016) [1], who studied non-ECAPed ZE41 Mg alloy. Based on experimental data, it is obvious that HVOF coatings frequently reveal a microhardness gradient along the coating thickness, with the maximum hardness near the interface and the lowest hardness at the substrate.

Due to a more uniform microstructure and fewer surface imperfections, the microhardness in the interface area is increased. Notably, when comparing Figure 4 and Figure 5, it becomes apparent that the microhardness of the SS316 coated non-ECAPed Mg alloy exhibits larger fluctuations compared to SS316 coated ECAPed Mg alloy. The observed changes in microhardness offer additional proof that the microstructure of SS316-coated ECAPed Mg alloy becomes more uniform and compact. These findings are consistent with previous studies conducted by Ma et al., (2008) and Sundararajan and Krishna (2003) [5,34]. Additionally, the variations in microhardness at the interface region of the HVOF-based SS316 coating on ECAPed AZ80 and AZ91 Mg alloys are linked to alterations in the microstructure, specifically grain refinement and the formation of secondary phases. As illustrated in Figure 3 and Figure 4, these modifications lead to a denser coating. The research also unveiled the feasibility of applying wear-resistant SS316 coatings onto ECAP-processed magnesium alloy substrates through HVOF. The kinetic energy of SS316 elements facilitated a “self-roughening” effect, enabling successful deposition with strong bonding.

### 3.3. Analysis of Corrosion Behavior of SS316-Coated AZ80/91 Mg Alloys

In this investigation, the HVOF technique was employed to apply coating on two magnesium alloys—AZ80 and AZ91. The coated specimens were evaluated concerning corrosion resistance and surface structure. The combined impact of ECAP and coating significantly improved corrosion behavior, elaborated upon in-depth. Figure 7 and Figure 8 depict the anodic–cathodic potentiodynamic polarization curves of AZ80 and AZ91 Mg alloys, both with and without ECAP treatment, after the SS316 coating process. These measurements were performed in a NaCl aqueous solution with a concentration of 3.5 wt.%. The point where the anodic and cathodic curves intersect indicates the corrosion potential (Ecorr) of the SS316 coating on both the non-ECAPed and ECAPed AZ80/91 Mg alloys. To facilitate comparison, the Ecorr value corresponding to the non-ECAPed AZ80/91 Mg alloy is also included in the plot.

The open-circuit corrosion potential of the non-ECAPed AZ80 and AZ91 Mg alloy specimens is measured at approximately −1.547 VSCE and −1.539 VSCE, respectively, and these values are observed to be highly similar to those of the ECAPed AZ80 and AZ91 Mg alloys. This indicates that the AZ80/91 Mg alloy substrate has low corrosion protection. The current density values of non-ECAPed, ECAP-2P, and ECAP-4P AZ80 specimens are measured to be 0.297 mA/cµ, 0.0295 mA/cm^2^, and 0.0084 mA/cm^2^, respectively. Similarly, for non-ECAPed, ECAP-2P, and ECAP-4P AZ91 specimens, the measured current density values are 0.263 mA/cm^2^, 0.0224 mA/cm^2^, and 0.0026 mA/cm^2^, respectively (as represented in Figure 9).

For AZ80 Mg alloy, the corrosion potentials (Ecorr) of 4P-ECAP/SS316, non-ECAPed/SS316, and 2P-ECAP/SS316 coatings are measured at −1.203 VSCE, −1.211 VSCE, and −1.206 VSCE, respectively. These results are similar to the Ecorr of SS316 bulk coating material (−1.2 VSCE). Additionally, all three coatings exhibit the lowest corrosion current densities, measuring 0.184 µA/cm^2^, 0.116 µA/cm^2^, and 0.108 µA/cm^2^, respectively (as shown in Figure 7). These studies demonstrate their usefulness as corrosion-protective coatings for Mg alloys, particularly at higher ECAP passes. The current densities for the uncoated non-ECAPed AZ80 Mg alloy (0.297 mA/cm^2^) are significantly larger, mainly due to the enhanced corrosion resistance of the SS316 material and the formation of a compact and dense coating on the substrate, as discussed in the previous section. Similar observations are made for AZ91 Mg alloys after SS316 coating, where the current densities are 0.136 µA/cm^2^, 0.120 µA/cm^2^, and 0.090 µA/cm^2^ for non-ECAPed/SS316, 2P-ECAP/SS316, and 4P-ECAP/SS316, respectively (as depicted in Figure 8).

Moreover, based on the polarization data acquired from the SS316-coated AZ80/91 Mg alloy specimens, it can be deduced that the SS-coated ECAP-4P AZ91 Mg alloy demonstrates superior resistance to corrosion compared to the SS316-coated ECAP-4P sample of AZ80 Mg alloy. Following the electrochemical polarization test, the corroded specimens were examined to assess the surface morphologies of the coating using SEM, as depicted in Figure 10 and Figure 11. The analysis reveals the occurrence of pitting corrosion zones, with noticeable defects such as pores and cracks underneath these regions, indicating that corrosion originated from these specific spots. Moreover, the corrosion resistance of the system can be notably enhanced with an SS coating. Additionally, utilizing an SS 316 coating can effectively prevent corrosion of magnesium substrates in environments with salt content.

Finally, Figure 12 depicts the difference in corrosion rates of AZ80/91 Mg alloy after ECAP-4P and SS316-coated ECAP-4P, indicating that SS316-coated ECAP-4P AZ80/91 alloy has higher corrosion resistance than uncoated ECAP-4P AZ80/91 alloy. This is because SS316 is more corrosion-resistant than the substrate material. Furthermore, ECAP-4P AZ91 Mg alloy is more corrosion resistant than ECAP-4P AZ80 Mg alloy. Figure 10 shows that the ECAP-4P sample dramatically enhanced corrosion resistance after SS316 coating, owing to SS316 being a more corrosion-resistant material than the substrate. The corrosion potential data for HVOF-based SS316-coated AZ80/91 Mg alloys, both non-ECAPed and ECAPed, are illustrated in Figure 9, where CR stands for “Corrosion Rate”, with units in millimeters per year (mm/y). The CR is determined using Equation (1).
(1)$$\mathrm{CR}\;(\mathrm{mm}/y)={3.27\times \;10}^{-3}\frac{i_{corr}\times A}{\rho }\mathrm{CR}$$

## 4. Discussion

Surface analysis of SS316-coated AZ80 and AZ91 Mg alloys involved SEM and EDS. Results in Figure 3 and Figure 4 show compact, uniform coatings (~80 ± 5 μm) with a well-deposited SS316 base layer, free from subsurface cracks or voids. Detailed microstructural analysis was previously presented [33]. Microhardness in HVOF-based SS316 coatings on AZ80/91 alloys varies, with the highest values near the interface and improvements seen in ECAPed samples. These findings align with prior research [1]. HVOF-coated AZ80 and AZ91 magnesium alloys underwent corrosion resistance evaluation. ECAP and coating improved corrosion resistance, seen in Figure 7 and Figure 8, showing potentiodynamic polarization curves. Non-ECAPed AZ80/91 alloys displayed low corrosion protection. ECAPed specimens, especially with more passes, exhibited lower current densities, making them effective corrosion-resistant coatings. Uncoated non-ECAPed AZ80 alloys had higher current densities due to SS316’s corrosion resistance and dense coating. AZ91 alloys showed similar trends post-SS316 coating. The study shows that SS316-coated ECAP-4P AZ91 Mg alloy is more corrosion resistant than AZ80 Mg alloy. SEM analysis reveals pitting corrosion zones and defects, while Figure 12 demonstrates higher corrosion resistance in SS316-coated ECAP-4P AZ80/91 compared to uncoated ECAP-4P AZ80/91. ECAP-4P AZ91 is more corrosion resistant than ECAP-4P AZ80. Figure 11 underscores that ECAP-4P samples significantly improve corrosion resistance after SS316 coating, primarily due to SS316’s superior corrosion resistance compared to the substrate.

## 5. Conclusions

AZ80/91 Mg alloy substrates are subjected to an equal-channel angular pressing (ECAP) process at 598 K using route R for improved corrosion characteristics. Then, additional improvement in the corrosion performance of AZ80/91 Mg alloy is achieved through the application of SS316 coating using the HVOF technique. The current research work yields the following findings:The microhardness of SS316 coating increases gradually from the substrate to the outermost layer. Notably, it reaches its maximum value as it approaches the interface zone. This can be attributed to the denser and more compact nature of the interface layer resulting from the HVOF technique compared to the inner and outer layers. As a consequence, the interface of SS316-coated ECAP-4P specimens for AZ80 and AZ91 Mg alloys increased in microhardness by 3.6% and 3.8%, respectively, as compared to the average microhardness of the SS316 coating.The application of an HVOF-based SS316 coating onto the non-ECAPed AZ80 magnesium alloy has demonstrated a remarkable enhancement in corrosion resistance. This improvement is evident in the substantial reduction in the corrosion current density, decreasing from 0.297 mA/cm^2^ to a mere 0.10 µA/cm^2^.The utilization of an HVOF-based SS316 coating on ECAPed AZ91 magnesium alloys led to a notable reduction in corrosion current density, diminishing it from 0.263 mA/cm^2^ to 0.090 mA/cm^2^. This outcome indicates that the synergistic combination of ECAP and SS coating holds promise as a viable strategy for enhancing both the mechanical and corrosion properties of magnesium alloys, particularly in automotive and aerospace applications.

## Figures and Tables

**Figure 1 materials-16-06651-f001:**
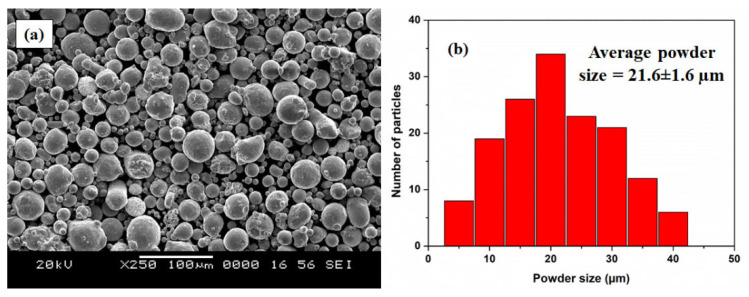
Experimental data of SS316 powder. (**a**) SEM image indicating variation in particle size, (**b**) graphical representation of particle size distribution.

**Figure 2 materials-16-06651-f002:**
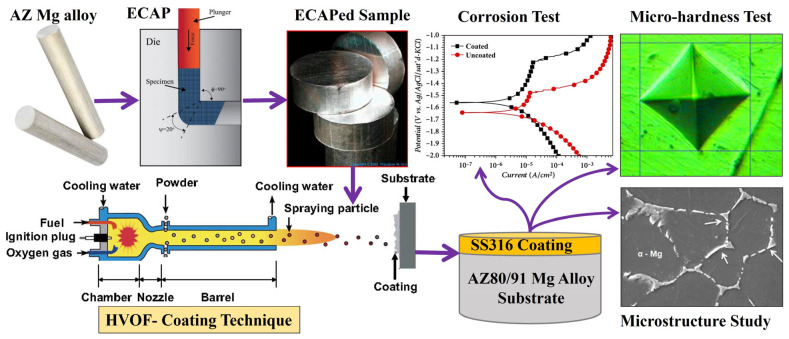
Experimental plan of current research work.

**Figure 3 materials-16-06651-f003:**
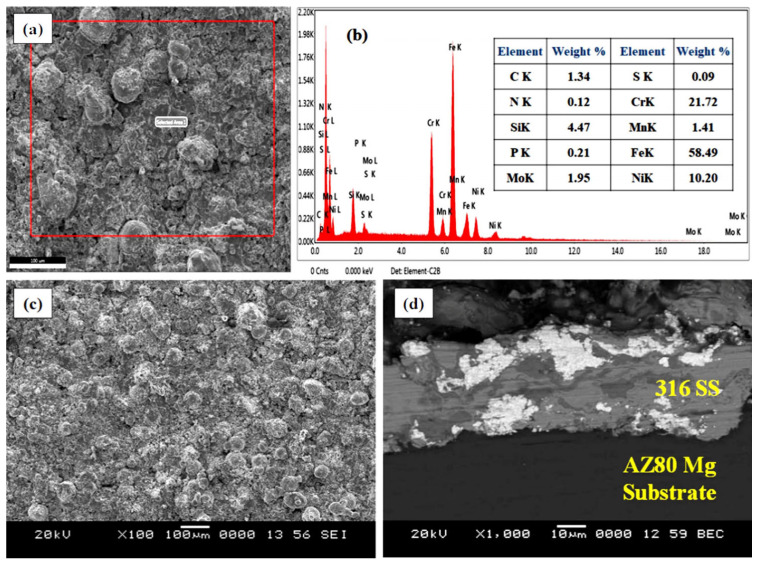
Microscopic and elemental analysis of HVOF-based SS316 coating formed on AZ80 Mg alloy. (**a**) SEM image of SS316 coating on non-ECAPed specimen; (**b**) EDS analysis of non-ECAPed specimen; (**c**) SEM image of SS316 on coating on ECAP-4P specimen; (**d**) interface region of SS316 coating and ECAP-4P specimen.

**Figure 4 materials-16-06651-f004:**
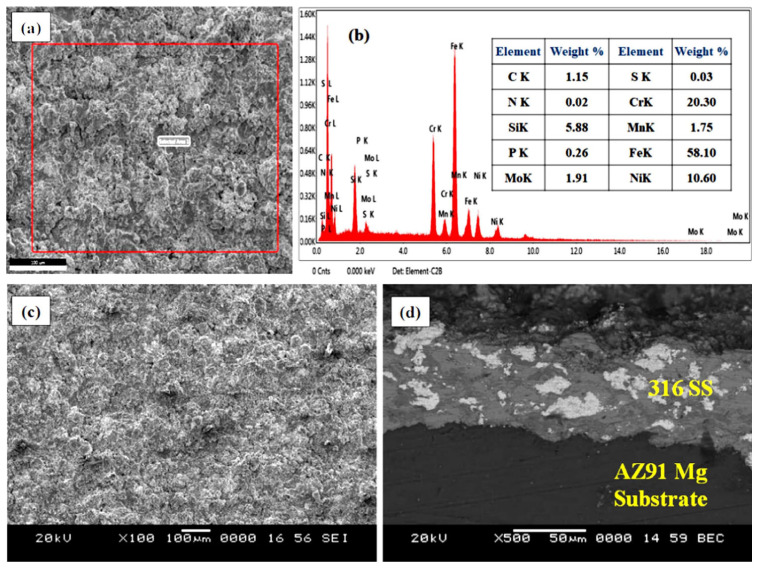
Microscopic and elemental analysis of HVOF-based SS316 coating formed on AZ91 Mg alloy. (**a**) SEM image of SS316 coating on non-ECAPed specimen; (**b**) EDS analysis of non-ECAPed specimen; (**c**) SEM image of SS316 on coating on ECAP-4P specimen; (**d**) interface region of SS316 coating and ECAP-4P specimen.

**Figure 5 materials-16-06651-f005:**
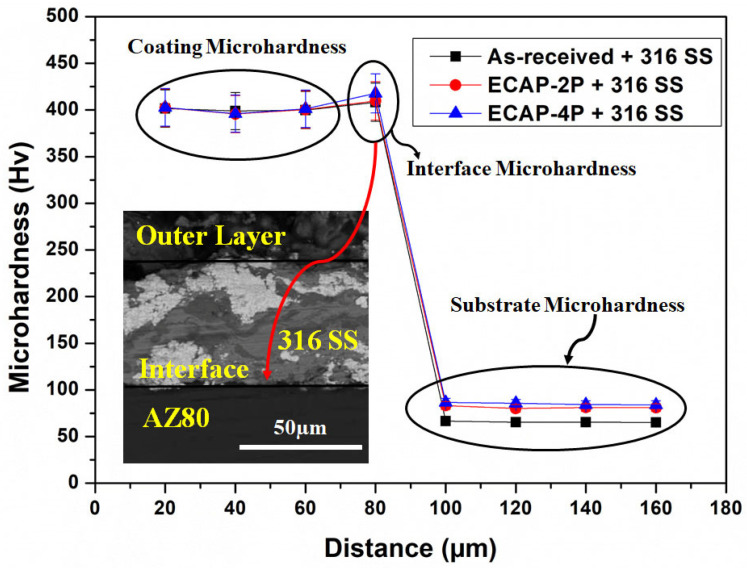
Sub-surface microhardness of SS316-coated AZ80 Mg alloys.

**Figure 6 materials-16-06651-f006:**
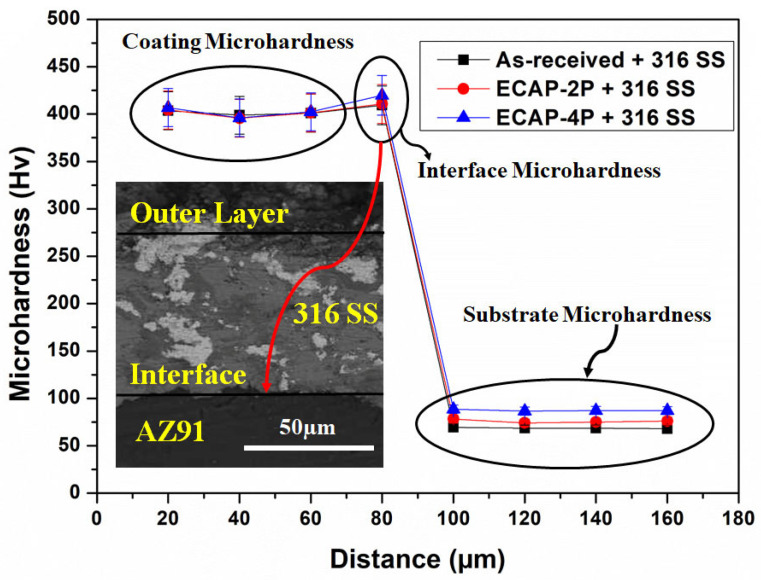
Sub-surface microhardness of SS316-coated AZ91 Mg alloys.

**Figure 7 materials-16-06651-f007:**
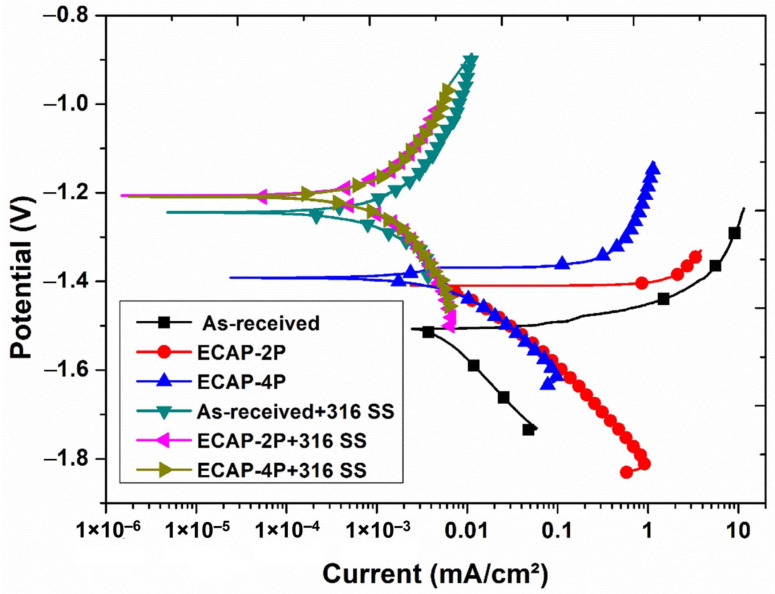
Graphical representation of corrosion behavior of HVOF-based SS316-coated ECAPed AZ80 Mg alloy.

**Figure 8 materials-16-06651-f008:**
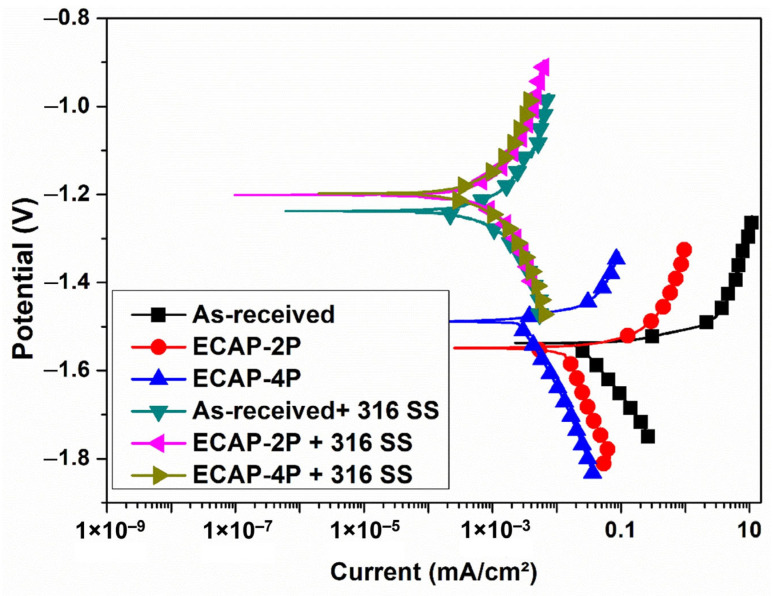
Graphical representation of corrosion behavior of HVOF-based SS316-coated ECAPed AZ91 Mg alloy.

**Figure 9 materials-16-06651-f009:**
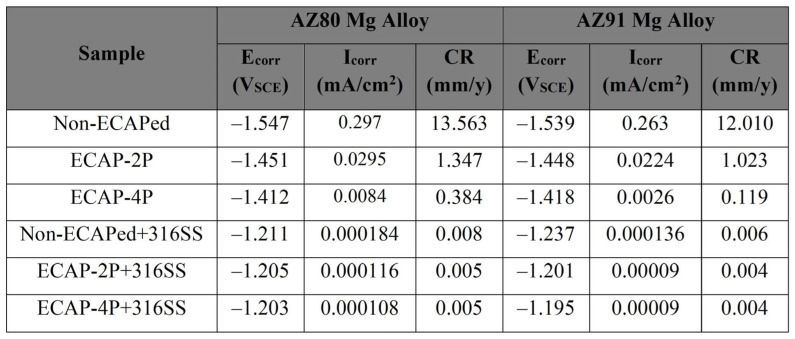
Corrosion potential data sheet corresponding to HVOF-based SS316-coated non-ECAPed/ECAPed AZ80/91 Mg alloy.

**Figure 10 materials-16-06651-f010:**
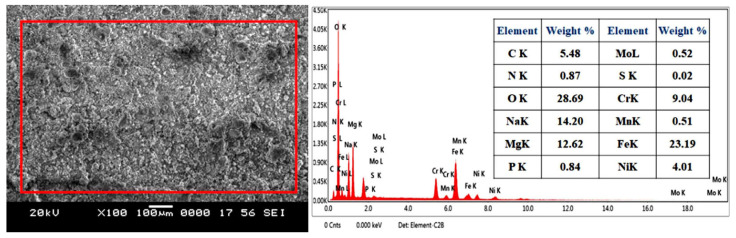
Corrosion morphology and EDS analysis of HVOF-based SS316-coated ECAP-4P AZ80 Mg alloys.

**Figure 11 materials-16-06651-f011:**
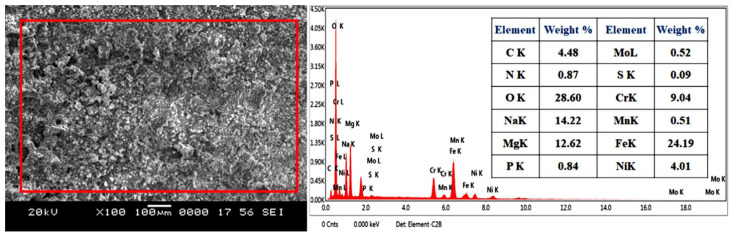
Corrosion morphology and EDS analysis of HVOF-based SS316-coated ECAP-4P AZ91 Mg alloys.

**Figure 12 materials-16-06651-f012:**
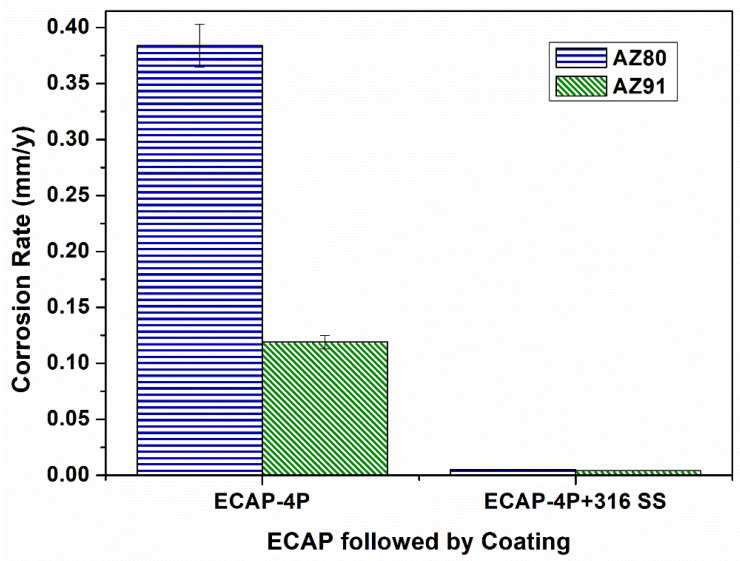
Graphical representation of corrosion rate of ECAP-4P AZ80/91 Mg alloy before and after the application of SS-316 coating.

**Table 1 materials-16-06651-t001:** Control parameters used during the HVOF spray coating process.

**Name of Substrate**	AZ80 and AZ91 Mg Alloy
**Type of Substrate**	ECAPed and non-ECAPed
**Coating material**	SS316 powder
**Size of coating material**	21.6 ± 1.6 μm
**Average coating thickness (µm)**	80–100
**Oxygen pressure (PSI)**	160–170
**Oxygen flow rate (SCFH)**	30–34
**Hydrogen pressure (PSI)**	120–140
**Fuel flow rate (SCFH)**	80–100
**Spraying distance (inch)**	8″–10″

## Data Availability

Not applicable.

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
