# Peer review of "Improvement in Corrosion Performance of ECAPed AZ80/91 Mg Alloys Using SS316 HVOF Coating"

_materials, 2023, doi:10.3390/ma16206651_

Round 1
Reviewer 1 Report
The paper reports about the improvement in corrosion resistence as a result of SS316 coatings. The system is interesting and the topic is suitable for Materials journal. However, several (minor) points must be addressed before publication.
1) The author must explain the acronym of HVOF before arriving to the end of the introduction
2) The aim of the work and the methods used should be better presented in the introduction. In the present version the introduction is quite confused.
3) In the Materials and Methods section the authors must report which kind of instruments were used in the experiments. The experimental conditions (for instance energy of electron beam, type of electrons (secondary, backscattered) used for the SEM images. Sample preparation methods
A revision of the text by a native english speaking is strongly recommended
Author Response
The author is really grateful to the reviewer for valuable insights and guidance in improving the manuscript. Your expertise greatly enriched the quality of our work.

Reviewer 2 Report
Comments of materials-2631783
In general, the manuscript is well written, however, there must be a more rigorous interpretation and scientific explanations of the mechanisms of the investigated parameters affecting the properties of Mg materials. The following comments must be addressed to add value to the manuscript:
1. The abstract can be improved to show more significance and findings of study.
2. 2. Experimental: The section pertaining to raw materials should be articulated in a more professional manner.
3. There are many methods to protect Mg alloys from corrosion, but the poor durability is critical for the large-scale applications. So, the authors should extend the test time to check the durability (OCP and EIS). And, the authors should compare their results with other's to show the advantage of the work.
4. I suggest that the test of neutral salt spray test should be added, because the test of corrosion resistance in the manuscript is too simple. It is not enough to only test the electrochemical performance.
5. It is suggested that the conclusion should be expressed more clearly and methodically.
Author Response

(The authors gave the same response as above.)

Reviewer 3 Report
The article entitled, "Improvement in Corrosion Performance of ECAPed AZ80/91 Mg alloys using SS316 HVOF Coating" discusses the improvements in corrosion resistance obtained for Mg alloys from the application of an inorganic, stainless steel coating. I noted some Technical Points that need to be addressed either by revising the text or adding explanations and a few minor points to correct typos.
Technical Points
1. What does "Notably, adhesion wear dominancy was observed [14]." mean?
2. For the paragraph that starts with, "In this present study, the corrosion resistance of Mg alloys is examined...", it is not clear if the authors are discussing their work, or work from the literature. Please provide a clear distinction that motivates the experimental work done in this paper from the previous literature.
3. In this sentence, "Notably, when HVOF sprayed nanostructured WC-10Co-4Cr coatings were exposed to seawater, they exhibited lower corrosion current density values, showcasing improved resistance to corrosion.", what are the corrosion currents lower than? What is this in comparison to?
4. This sentence is unclear, "While there have been various studies on HVOF coatings on magnesium and its alloys, however the specific corrosion behavior of HVOF-based SS316-coated ECAPed AZ80/91 Mg alloys are hardly reported so far for automotive and aerospace application." In addition, the previous points have all been discussing WC10Co4Cr or Cr3C2-NiCr coatings, but in this sentence the SS316 coating is introduced. What are the differences among the 3 coatings? Are the conclusions and findings from the other coatings relevant for SS316 coatings? This relationship could be clarified.
5. Can the authors clarify what are the comparisons for "more compact" and "more uniform" in this sentence: "A microscopic investigation of SS316-coated surfaces of the AZ80 and AZ91 Mg alloy reveals more compact and uniform coating as illustrated in Figure 3 and Figure 4 in case of both ECAPed and non-ECAPed condition."
6. Is this statement supported by the data? "It is important to note that the ECAPed samples with SS316 coating exhibited a significant improvement in microhardness compared to non-ECAPed SS316- coated AZ80/91 Mg alloy coated with 316 SS as shown in Figure 5 and Figure 6." It appears to me that the SS316 coatings have very similar hardness levels while the ECAP treatment only changes the hardness of the substrate material by 10-15 Hv for the AZ-80 alloy and even less than that for the AZ-91 alloy?
7. The polarization curves shown in Figures 7 and 9 need more interpretation. The open-circuit values for the stainless-steel clad alloys is very low compared to the expected open-circuit potential for 316. This suggests the OCP is due to a mixed potential between the stainless-steel coating and the Mg alloy substrate. However, that means the coating has through-thickness defects. Is that the case? Or, is there another explanation for the low OCP?
8. A problem with determining corrosion rates from polarization curve data, is that it assumes the conditions obtained during the polarization experiment will remain constant. However, that is usually not the case. One way to improve the comparison of the corrosion behaviors of the different alloys would be to monitor the corrosion current for freely-corroding samples over the course of 24 - 48 h to see if the corrosion currents remain constant or decay over time.
9. The first statement in the Conclusions is not supported by the details provided in the text. "AZ80/91 Mg alloy substrates are subjected to Equal-channel-angular-pressing 279 (ECAP) process at 598K using route R for improved mechanical characteristics."
Minor Points
1. Acronym for HVOF not defined near its first use.
2. Cermet coating not defined.
3. What do ECAP-2P and ECAP-4P mean?
Moderate edits needed on a few sentences that were confusingly written that were noted in the Author comments.
Author Response

(The authors gave the same response as above.)

Reviewer 4 Report
Magnesium and its alloys are very popular due to their light weight, good machinability and relatively good mechanical properties. In order to expand their use in various fields of industry, it is necessary to improve their corrosion resistance.
The article submitted to me for review is structured correctly. Contains all parts corresponding to scientific research. Many abbreviations are used, and it is necessary to enter them the first time they are written in the text. For example, the description of the HVOF technique should be presented at the beginning of the Introduction section. In order to increase the interest of the readers, I would suggest giving the chemical composition of the substrate in the Materials and Methods section, regardless of the fact that the alloys are known. In the same section, it is necessary to describe the physical and electrochemical methods for determining corrosion resistance, as well as why exactly they were chosen. Energy-dispersive X-ray spectroscopy is represented by the abbreviation EDS and XEDS - it is necessary to unify. It is accepted to denote the method by EDS or EDX. In the electrochemical tests for determining the corrosion resistance, the Ecorr parameter is presented. How is it defined? It is not determined by the intersection of the anode and cathode curves. If the authors defined it as such, then its value is not quite correct. How is the open circuit potential determined given that the authors work in potentiodynamic mode? What is the scan rate of the potential to judge whether the two values ​​are close or not? What do the authors mean by the abbreviation CR mm/y presented in fig. 12, which should be presented as a table? If it is permeability, how is it determined or calculated? It is necessary to correct the inaccuracies in the dimensions.
The English language needs a little checking for grammatical and stylistic errors.
Author Response

(The authors gave the same response as above.)

Round 2
Reviewer 3 Report
The revised version of the manuscript entitled, "Improvement in Corrosion Performance of ECAPed AZ80/91 3 Mg alloys using SS316 HVOF Coating" has addressed the reviewers' technical concerns. I believe I have a better sense of where this work fits in the field and the significance of the observations. There are still a few typo mistakes, such as the use of "oxy" instead of "oxygen" and the order of paragraphs in the Materials and Methods section seems strange. However, I think the copy editor will be able to address those points better than I can.
See notes in the Authors' Suggestions.